

# Mediterranean Sea Hydrographic Atlas: towards optimal data analysis by including time-dependent statistical parameters

Athanasia Iona[1,2], Athanasios Theodorou[2], Sylvain Watelet[3], Charles Troupin[3], and Jean-Marie Beckers[3]

[1]Hellenic Centre for Marine Research, Institute of Oceanography, Hellenic National Oceanographic Data Centre, 46,7 km Athens Sounio, Mavro Lithari P.O. BOX 712 19013 Anavissos, Attica, Greece
[2]University of Thessaly, Department of Ichthyology & Aquatic Environment, Laboratory of Oceanography, Fytoko Street, 38 445, Nea Ionia Magnesia, Greece
[3]University of Liège, GeoHydrodynamics and Environment Research, Quartier Agora, Allée du 6-Août, 17, Sart Tilman, 4000 Liège 1, Belgium

**Correspondence:** Athanasia Iona (sissy@hnodc.hcmr.gr)

**Abstract.** The goal of the present work is to provide the scientific community with a high-resolution Atlas of temperature and salinity for the Mediterranean Sea based on the most recent datasets available and contribute to the studies of the long-term variability in the region. Data from the Pan-European Marine Data Infrastructure SeaDataNet were used, the most complete and, to our best knowledge, of best quality dataset for the Mediterranean Sea as of today. The dataset is based on in situ measurements acquired between 1900–2015. The Atlas consists of horizontal gridded fields produced by the Data Interpolating Variational Analysis, where unevenly spatial distributed measurements were interpolated onto a $1/8° \times 1/8°$ regular grid on 31 depth levels. Seven different types of climatological fields were prepared with different temporal integration of observations. Monthly, seasonal and annual climatological fields have been calculated for all the available years, seasonal to annual climatologies for overlapping decades and specific periods. The seasonal and decadal time frames have been chosen in accordance with the regional variability and in coherence with atmospheric indices. The decadal and specific periods analysis was not extended to monthly resolution due to the lack of data, especially for the salinity. The Data Interpolating Variational Analysis software has been used in the Mediterranean Region for the SeaDataNet and its predecessor Medar/Medatlas Climatologies. In the present study, a more advanced optimization of the analysis parameters was performed in order to produce more detailed results. The Mediterranean Region past and present states have been extensively studied and documented in a series of publications. The purpose of this Atlas is to contribute to these climatological studies and get a better understanding of the variability on time scales from month to decades and longer. Our gridded fields provide a valuable complementary source of knowledge in regions where measurements are scarce, especially in critical areas of interest such as the Marine Strategy Framework Directive (MSFD) regions. The dataset used for the preparation of the Atlas is available from https://doi.org/10.12770/8c3bd19b-9687-429c-a232-48b10478581c.



## 1 Introduction

In oceanography, a climatology is defined as a set of gridded fields that describe the mean state of the ocean properties over a given time period. It is constructed by the analysis of in-situ historical data sets and has many applications such as initialization of numerical models, quality control of observational data in real time and delayed mode, and it is used as a baseline for

comparison to understand how the ocean is changing. The Mediterranean Sea is among the most interesting regions in the world because it influences the global thermohaline circulation and plays an important role in regulating the global climate (Lozier et al., 1995; Béthoux et al., 1998; Rahmstorf, 1998). It is therefore essential to improve our understanding of its dynamics and its process variability. It is a semi-enclosed sea divided by the Sicily Strait in two geographical basins: the Western Mediterranean and the Eastern Mediterranean and characterized by peculiar topographic deep depressions where nutrient-rich deep-water

masses are stored for long time (Manca et al., 2004). It is a concentration basin, where evaporation exceeds precipitation. The general mean circulation pattern is driven by the continuous evaporation, heat fluxes exchanges with the atmosphere, the wind stress and the water masses exchanges between its basins and sub basins, as described in Robinson et al. (2001). In the surface layer (0–150 m) there is an inflow of warm and relatively fresh Atlantic water (AW; S $\approx$ 36.5, T $\approx$ 15°C) which is modified along its path to the Eastern basin following a general cyclonic circulation. The intermediate layers (150–600 m) are dominated

by the saline Levantine Intermediate Water (LIW; S $\approx$ 38.4, T $\approx$ 13.5°C), regularly formed at the Levantine basin. LIW is one of the most important Mediterranean water masses because it constitutes the higher percentage of the outflow from the Gibraltar Strait towards the Atlantic Ocean. In the deep layers, there are two main thermohaline cells. Deep waters formed via convective events in the northern regions of the Western Mediterranean (WMDW-Western Mediterranean Deep Water; S $\approx$ 38.44–38.46, T $\approx$ 12.75–13.80°C) at the Gulf of Lions and at the north regions of the Eastern Mediterranean (EMDW-Eastern

Mediterranean Deep Waters) at the Adriatic (S $\approx$ 38.65, T $\approx$ 13.0°C) and Cretan Sea ($\sigma_\theta < 29.2$). On top of this large scale general pattern are superimposed several sub-basin scale and mesoscale cyclonic and anticyclonic motions due to topographic constraints and internal processes. The first climatological studies of the Mediterranean go back to 1966 with Ovchinnikov who carried out a geostrophic analysis to compute the surface circulation. The circulation features were describing a linear, stationary ocean. In the 1980s and 1990s, through a comprehensive series of observational studies and experiments in the

Western Mediterranean (La Violette, 1990; Millot, 1987, 1991, 1999), the sub-basin and mesoscale patterns were discovered and the crucial role of eddies in modifying the mean climatological circulation and mixing properties inside the different sub-basins which include the Tyrrhenian, Ligurian and Alboran Seas was emphasized (Bergamasco and Malanotte-Rizzoli, 2010). In 1987, Guibout constructed charts with the typical structures observed at sea, but with limited climatological characteristic as this atlas was based on the quasi-synoptic information of the selected cruise used. In 1982, an international research group was

formed, called POEM (Physical Oceanography of the Eastern Mediterranean, 1984), under the auspices of the IOC/UNESCO and of CIESM (Commission Internationale pour l'Exploration Scientifique de la Méditerranée) focused on the description of the phenomenology of the Eastern Mediterranean, by analysing historical data and collecting new data (Malanotte-Rizzoli and Hecht, 1988). However the focus was on the Eastern Mediterranean Sea, as there was very little knowledge of this basin compared to the Western Mediterranean and other world regions and still there was missing a global data set to describe the





eastern Mediterranean efficiently and its interaction with the western part. Hecht et al. (1988), by analyzing hydrographic measurements in the south eastern Levantine Basin, succeeded to describe the climatological water masses of the region and identify their seasonal variations. In the western Mediterranean, Picco (1990) conducted an important climatological study and constructed a climatological atlas analyzing around 15000 hydrological profiles from various sources for the period 1909–

5 1987. In 1982, Levitus published the first climatological atlas of the world ocean at a 1° resolution, using temperature, salinity and oxygen data from CTD, Bottle stations, mechanical and expendable bathythermographs of the previous eighty years. An objective analysis at standard levels was followed to compute the climatology. Since 1994, a new version is released every 4 years. The current version that includes the Mediterranean (World Ocean Atlas 2013, V2) (Locarnini et al., 2013) is a long-term set of objectively analyzed climatologies of temperature, salinity, oxygen, phosphate, silicate, and nitrate for annual, seasonal,

and monthly periods for the world ocean. It also includes associated statistical fields of observed oceanographic profile data from which the climatologies were computed. In addition to increased vertical resolution, the 2013 version has both 1° and 1/4° horizontal resolution versions available for annual and seasonal temperature and salinity for six decades, as well as monthly for the decadal average. Brasseur et al. (1996) introduced a new method to reconstruct the three-dimensional fields of the properties of the Mediterranean Sea. Seasonal and monthly fields were analyzed using a Variational Inverse Method (VIM) to

generate the climatological maps, instead of the objective analysis introduced by Gandin (1966); Bretherton et al. (1976) to the meteorology and oceanography and widely used by then. More than 34000 CTD and Bottle data were used and integrated in the so-called MED2 historical data base. Comparison of the results obtained with both methods showed that VIM was mathematically equivalent but numerically more efficient than the objective analysis. The method has been adopted by the Medar/Medatlas and its successor SeaDataNet Project.

At the beginning of the 2000s, many research projects and monitoring activities have produced large amounts of multidisciplinary in-situ hydrographic and bio-chemical data for the whole Mediterranean Sea but still the data were fragmented and inaccessible from the scientific community. The aim of the EU/MAST Medar/Medatlas Project, 2001 (http://www.ifremer.fr/medar/) was to rescue and archive the dispersed data through a wide cooperation of countries and produce an atlas for 12 core parameters: temperature and salinity, dissolved oxygen, hydrogen sulphur, alkalinity, phosphate, ammonium, nitrite, ni-

trate, silicate, chlorophyll and PH. Gridded fields have been computed on a 1/4° grid resolution using the Variational Inverse Method (VIM) and the DIVA tool developed by the GHER group of the University of Liège. Inter-annual and decadal variabilities of temperature and salinity were computed as well. The atlas is available at http://modb.oce.ulg.ac.be/backup/medar/contribution.html and on CD-Rom (MEDAR Group, 2002). The atlas contains a selection of figures and 3D fields in netCDF. The EU/FP7 SeaDataNet Project 2006-2011, 2012-2016 (the successor of the EU/MAST Medar/Medatlas) has integrated

historical, multidisciplinary data on a unique, standardized online data management infrastructure and provides value-added aggregated datasets and regional climatologies based on these aggregated datasets for all the European sea basins. SeaDataNet has adopted the VIM method and the DIVA software tool. Temperature and salinity monthly climatologies have been produced on a 1/8° grid resolution. These climatologies are based on the V1.1 historical data collection of publicly available temperature and salinity in situ profiles (http://dx.doi.org/10.12770/8c3bd19b-9687-429c-a232-48b10478581c) spanning the time period 1900-2013 (Simoncelli et al., 2015).



## 1.1 Objectives

The objective of this study is the computation of an improved Atlas compared with the existing products in the Mediterranean
Sea using the latest developments of the DIVA tool with the aim to contribute to the better representation of the climatological
patterns and understanding of the long-term variability of the regional features of the basin. The originality of this product com-
pared to the SeaDataNet climatology (both products use similar techniques) is the higher temporal resolution, up to decadal,
and more advanced calibration of the analysis parameters for improving the results and the representation of general circulation
patterns at time scales smaller than the climatic means. Besides the WOA13 decadal periods climatologies (1955–64, 1965–
74, 1975–84, 1985–94, 1995–2004 and 2005–2012) the Medar/Medatlas provides interannual and decadal gridded fields and
therefore it is compared with the present Atlas. The present product has two major improvements: higher spatial and vertical
resolution and error fields that accompany all the analysis results, allowing therefore a more reliable assessment of the results.
The advantage of this product in relation to the WOA13 climatology is the higher spatial resolution up to $1/8° \times 1/8°$ longi-
tude/latitude grids used and the higher temporal resolution as the analysis uses running decades instead of successive decades.
Another important difference with respect to all previous existing climatologies in the Mediterranean is that additional and
more recent data are used. The Atlas constructed consists of climatological fields (called climatology hereafter) to depict the
"mean" state of the Mediterranean region at monthly, seasonal: winter (Jan–Mar), spring (Apr–Jun), summer (Jul–Sep), fall
(Oct–Dec), and annual scale. Additionally, climatological fields at seasonal and annual scale for 57 running decades have been
produced to depict the interannual and decadal variability of the system and reveal the decadal trends.

One additional period from 2000–2015 was produced for those users or applications who are interested to reference to the
latest data and new platform types such as Argo floats. This period is provided with the five previous successive decadal periods
e.g. 1950–1959, 1960–1969, 1970–1979, 1980–1989, 1990–1999 and 2000–2015. In summary, the following gridded products
are available:

- Monthly climatological gridded fields obtained by analyzing all data of the whole period 1950 to 2015 falling within
  each month.

- Seasonal climatological gridded obtained by analyzing all data of the whole period from 1950 to 2015 falling within
  each season.

- Seasonal gridded fields obtained by analyzing all data falling within each season for each of 57 running decade from
  1950–1959 to 2006–2015.

- Seasonal gridded fields obtained by analyzing data falling within each season for six periods: 1950–1959, 1960–1969,
  1970–1979, 1980–1989, 1990–1999, 2000–2015.

- Annual climatology obtained by analyzing all data (regardless month or season) for the whole period from 1950 to 2015.

- Annual gridded fields by analyzing all data regardless month or season for each of the 57 running decade from 1950–
  1959 to 2006–2015.



– Annual gridded fields obtained by analyzing data falling with each season for six periods, 1950–1959, 1960–1969, 1970–1979, 1980–1989-1990–1999, 2000–2015. The first five periods coincide with the corresponding decades.

The Atlas covers the geographical region -6.25°W–36.5°E, 30°–46°N on 31 standard depth levels from 0–4000m. The fields are stored in netCDF files. The Atlas is accessible in netCDF from the Zenodo platform using the following DOIs:

**Annual Climatology:** https://doi.org/10.5281/zenodo.1146976.

**Seasonal Climatology for 57 running decades**  from 1950-1959 to 2006-2015:
    https://doi.org/10.5281/zenodo.1146938.

**Seasonal Climatology:**  https://doi.org/10.5281/zenodo.1146953.

**Annual Climatology for 57 running decades**  from 1950-1959 to 2006-2015:
    https://doi.org/10.5281/zenodo.1146957.

**Seasonal Climatology for six periods:**  1950-1959, 1960-1969, 1970-1979, 1980-1989, 1990-1999, 2000-2015:
    https://doi.org/10.5281/zenodo.1146966.

**Annual Climatology for six periods:**  1950-1959, 1960-1969, 1970-1979, 1980-1989, 1990-1999, 2000-2015:
    https://doi.org/10.5281/zenodo.1146970.

**Monthly Climatology:**  https://doi.org/10.5281/zenodo.1146974.

Tables 1 and 2 give the state of the art of the existing climatologies in the Mediterranean Sea.

## 2    Data

### 2.1    Data source

The SeaDataNet Temperature and Salinity historical data collection V2 for the Mediterranean Sea was used (INVG, 2015, http://sextant.ifremer.fr/record/8c3bd19b-9687-429c-a232-48b10478581c/). The collection includes 213542 temperature and 138691 salinity profiles from in-situ measurements covering the 1911–2015 period, which corresponds to all open access data available through the European SeaDataNet Marine Data Infrastructure (www.seadatanet.org). These datasets were collected
from 102 data providers, are quality controlled and archived in 32 marine and oceanographic data centres and distributed into the infrastructure by 27 SeaDataNet partners. The data range from 9.25°W to 37°E and include a part of the Atlantic (not included in the Atlas) and the Marmara Sea. Figure 1 shows the locations of the profiles and Table 3 the main instruments of the collection.

The profiles in the SeaDataNet collection come from in-situ observations collected with various instruments and platforms such as CTD and Bottles data by discrete water samplers operated by research or other smaller vessels, bathythermographs



**Table 1.** Overview of characteristics of existing T/S climatologies in Mediterranean Sea.

| Climatology | World Ocean Atlas 2013 | Medar/Medatlas | SeaDataNet | Present |
|---|---|---|---|---|
| **Date** | 2013 | 2002 | 2015 | 2017 |
| **Instruments/platforms** | Bathythermographs, discrete water samplers, CTD, drifters, profiling floats, gliders | Bathythermographs, discrete water samplers, CTD, thermistor chains | Bathythermographs, discrete water samplers, CTD, thermistor chains, thermos-alinographs, drifters, profiling floats, moorings | Bathythermographs, discrete water samplers, CTD, thermistor chains, thermos-alinographs, drifters, profiling floats, moorings |
| **Horizontal extent** | Global | Mediterranean, Black Sea | Mediterranean Sea | Mediterranean Sea |
| **Parameters** | Temperature, salinity | Temperature, salinity, bio-chemical parameters | Temperature, salinity | Temperature, salinity |
| **Horizontal resolution** | 1/4° × 1/4° | 1/4° × 1/4° | 1/8° × 1/8° | 1/8° × 1/8° |
| **Vertical extent** | 0–1500 m | 0–4000 m | 0–5500 m | 0–5500 m |
| **Vertical levels** | 57 | 25 | 33 | 33 |
| **Temporal data coverage** | 1864–2013 | 1890–2000 | 1900–2013 | 1900–2015 |
| **Temporal resolution** | Climatic, monthly, seasonal, averaged periods | Climatic, monthly, seasonal, inter-annual*, running 3-years*, running 5-years, running decades* | Monthly | Climatic, monthly, seasonal, running decades, averaged periods |

No error fields are available for the Medar/Medatlas, inter-annual*, running 3-years* and running decades* computations.

(mechanical or expendable). Table 3 summarizes the major numbers for temperature and salinity profiles and their temporal spanning as derived from the station type, the instrument and the platform type.

5    There are 21682 vertical profiles of Temperature and Salinity from CTD and Bottle stations with no information on the instrument used. These figures correspond to profiles before the quality control and the processing of data for preparation of climatology.

The Bathythermographs data were maintained in the collection despite the incorrect fall rate and the resulting warm bias in the measurements (Wijffels et al., 2008). Yet, there is not a XBT/MBT correction for the Mediterranean as for the global ocean (https://www.nodc.noaa.gov/OC5/XBT_BIAS/xbt_bias.html). These data improve significantly the geographical coverage of





**Table 2.** Analysis parameters for Variational Inverse Method (VIM) implementation in the Mediterranean.

| Climatology | Medar/Medatlas | SeaDataNet | Present |
|---|---|---|---|
| **Method** | VIM | DIVA (VIM) | DIVA (VIM) |
| **Correlation length** | constant | constant | variable |
| **Signal-to-noise ratio** | variable | constant | variable |
| **Background field** | semi-normed | semi-normed | semi-normed |
| **Detrending** | No | No | Yes (for background fields) |
| **Observation weighting** | No | No | Yes |

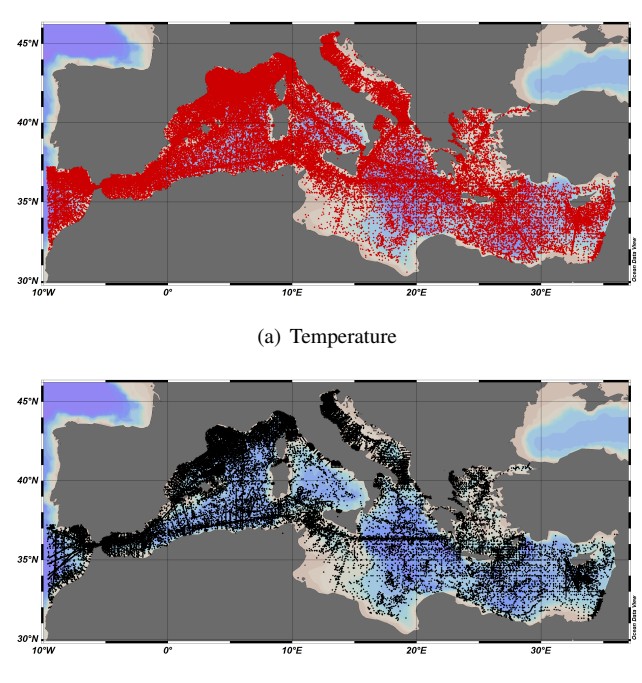

(a) Temperature

(b) Salinity

**Figure 1.** Geographical distribution of (a) temperature and (b) salinity profiles of the SeaDataNet V2 historical data collection.

the temperature records during the 1950–1985 period and reduce the resulting analysis error due to data gaps. Therefore, it was decided to keep them despite their known (small) bias.

5 **2.1.1 Preparation for the analysis**

The following operations were applied to the initial data before performing the analysis:

1. Data were extracted at the selected depth levels and temporal frames (months, seasons, decades and periods).



**Table 3.** Data distribution per instrument, station and platform type.

| | | Number of profiles | | |
| | | Temperature | Salinity | Period |
| **Instrument/Station Type** | Platform type | | | |
|---|---|---|---|---|
| **CTD** | Research vessel, moored surface buoy, fixed mooring, ships of opportunity, other type of vessel | 45914 | 42420 | 1964–2013 |
| **Discrete water samplers** | Research vessel, other type of vessels | 33449 | 30423 | 1912–2009 |
| **Bathythermographs** | Research vessel, ships of opportunity, other type of vessels | 44336 | | 1952–2014 |
| **Thermosalinographs** | Research vessel, other type of vessels | 51063 | 29747 | 2008–2009 |
| **Salinity sensors, water temperature sensor** | Drifting subsurface float | 8206 | 8207 | 2005–2015 |
| **Not provided** | Research vessel, ships of opportunity, other type of vessel, unknown | 21682 | 21682 | 1914–2013 |

2. Spatial binning was performed to improve the quality of the parameters (correlation length and signal-to-noise ratio) optimization by averaging a part of the highly correlated observations (taken by a research vessel in a restricted area, for instance). Spatial binning was applied to the data only during the parameters optimization step but then the original data (and non-binned) were used for the analysis itself.

3. Data outside the analysis domain were excluded.

4. Weights were applied to the data to reduce the influence of large number of data located in a small area within a short period. This is particularly useful for time series data. Characteristic length of weighting was set equal to 0.08° and characteristic time of weighting was set equal to 90 days for the seasonal analysis and 30 days for the monthly analysis. The scales were chosen according to the spatial and temporal resolution of the analysis.

### 2.1.2 Vertical interpolation

The observations do not have a uniform vertical distribution and a vertical interpolation of the profiles into standard depths is needed prior to the gridding. DIVA tool has embedded the weighted parabolic interpolation method (Reiniger and Ross, 1968) into its workflow (Troupin et al., 2010). This method is widely used in oceanographic climatologies, such as World Ocean Atlas 2013 (WOA13) (Locarnini et al., 2013; Zweng et al., 2013), Medar/Medatlas (MEDAR Group, 2002), as it creates less vertical instabilities. Only "good" data were used, e.g. data with quality control flag 1 and 2 according to the SeaDataNet QC flag scale (SeaDataNet Group, 2010). The interpolation was performed onto 31 IODE standards depths [0, 5, 10, 20, 30, 50, 75, 100, 125, 150, 200, 250, 300, 400, 500, 600, 700, 800, 900, 1000, 1100, 1200, 1300, 1400, 1500, 1750, 2000, 2500, 3000, 3500, 4000]. These are the same depths as those used in the existing SeaDataNet climatology, thus allowing quick visual



intercomparisons. The WOA13 climatological fields offer a higher vertical resolution (87 depth levels from 0–4000 m) thus
facilitating higher resolution models or more accurate quality control for observational data. However, the monthly WOA13
fields in the Mediterranean extend only down to 1500 m. In the next releases of the present Atlas, the vertical resolution will
be increased. This version rather focus on the improvement of the horizontal and temporal resolutions.

### 2.1.3 Quality Control

Currently the SeaDataNet V2 collection is the most complete and quality controlled data set for the Mediterranean Sea. Re-
gional experts (INGV, 2015) responsible for the preparation of the climatology are performing the checks in close cooperation
with the data originators and the responsible data centres ensuring the best result. Prior to its release, it has undergone extended
quality control according to the SeaDataNet standards and strategy such as elimination of duplicates, broad range checks for
detecting outliers, spikes, density inversion, zero values (SeaDataNet Group, 2010). The quality control is done with the use of
Ocean Data View tool (ODV, Schlitzer, 2002, https://odv.awi.de). However, in order to avoid the influence of extremes (but not
necessarily erroneous values) in the climatology, the following additional checks were applied to the data. The Mediterranean
Sea was divided in 27 rectangular areas defined by Manca et al. (2004) which exhibit the typical sub-regional features. In each
area, a mean profile was calculated at the standard depths. Values laying outside $\pm$ 3 standard deviations around the mean were
considered as "outliers" and excluded from the DIVA analysis. The percentage of data excluded from the analysis was 0.8%
for the temperature and 0.9% for the salinity values. This filter with the standard deviation was applied twice at the data sets
that were used as background fields and for the optimization of the analysis parameters (the correlation length, signal-to-noise
ratio and variance of the background field). In the second run of the standard deviation filter, the amount of excluded data was
0.5% and 0.7% respectively.

### 2.1.4 Data temporal distributions

There is an expected seasonal distribution with more temperature and salinity profiles during summer and autumn as well as at
the surface layers compared to the deeper ones (Fig. 2). However the monthly and yearly distributions reveal a great bias for
September and October and for the years 2008 and 2009. This is due to the temporary presence of 51063 thermosalinograph
data, at 2 and 3  m depth, essentially located near the Strait of Gibraltar, and in the Alboran Sea. Such irregular distribution in
time has to be taken into account when computing the climatological fields otherwise the spatial interpolations will be biased
towards the values of these high data coverage during these two months. Therefore, a detrending method was applied to the
data prior to the DIVA analysis to remove this effect of uneven distribution in time. The detrending tool is provided by the
DIVA software. The detrending was applied to all background fields (see paragraph below) and the monthly, seasonal, and
annual climatologies. It was not applied to running decades and the 6 decadal period climatologies in order not to remove any
of the long term trends in temperature and salinity.



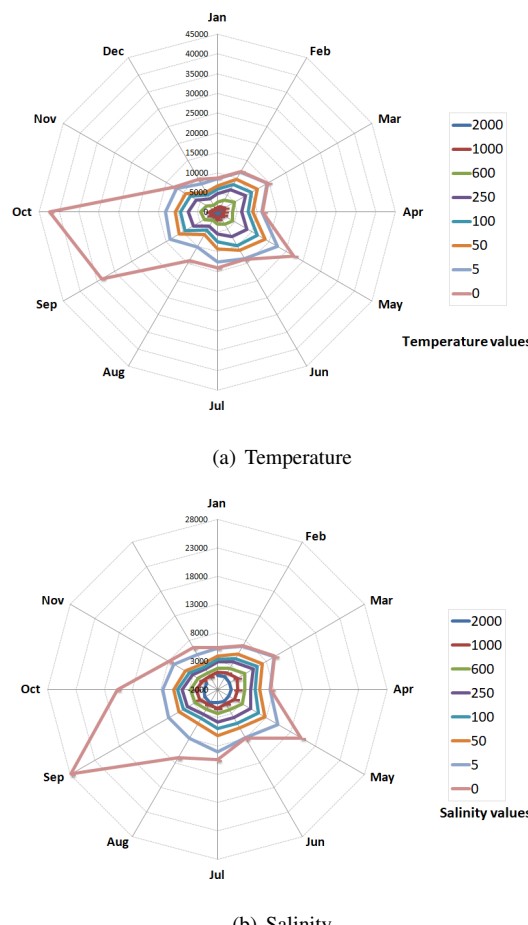

(a) Temperature

(b) Salinity

**Figure 2.** Number of observations per month for (a) temperature and (b) salinity.

### 2.1.5  Data weighting

The influence of the uneven distribution in space where a large number of data points are concentrated in a very small area and
within a very short period is controlled by applying different weights to each of these data points. Indeed such points cannot be
considered independent in a climatological analysis. So rather than to calculate super observation (similarly to binning), one
can reduce the weight or in other words, increase the error attached to each individual measurement. Points which are close in
time and space will undergo such a treatment. The scales below which such a weighting is done have a characteristic length of
0.08 degrees (same unit as the data locations) and a characteristic time of one month (30 days). Typical examples where the
weighting is particularly profitable are the cases of the thermosalinographs data (see previous paragraph) and time series data
from coastal monitoring stations. Data weighting functionality is also provided by the DIVA tool. Both data weighting and
detrending techniques have been applied for the first time in the computations for the Mediterranean Sea Climatologies.





## 3   Method

### 3.1   The DIVA interpolation tool

The Data Interpolating Variational Analysis (DIVA) is a method designed to perform spatial interpolation (analysis) of sparse
and heterogeneously distributed and noisy data into a regular grid in an optimal way. The basic idea of the variational analysis
is to determine a continuous filed approximating data and exhibiting small spatial variations. In other words, the target of
the analysis is defined as the smoothest fields that respects the consistency with the observations and a priori knowledge of
the background field over the domain of interest. To do so, a cost function that takes into account the distance between the
reconstructed field and the observations and the regularity of the field is minimised. The solution of the minimisation problem
is obtained through a finite-element technique (Rixen et al., 2000). The main advantage is that the computational cost is
independent of the number of data analyzed, instead it depends on the number of degrees of freedom i.e. on the size of the
finite-element mesh. The mesh takes into account the complexity of the geometry of the domain without having to separate sub-
basins prior to the interpolation and automatically prohibiting correlations across land barriers. Among other major advantages
of the method, DIVA can take into account dynamic constraints allowing for anisotropic spatial correlation. Tools to generate
the finite element mesh are provided as well as tools to optimize the parameters of the analysis. The signal-to-noise ratio is
optimized by a Generalized Cross Validation (GCV) technique (Brankart and Brasseur, 1996) while the correlation length is
estimated by comparing the relation between the empirical data covariance and the distance against its theoretical counterpart
(Troupin et al., 2017). Along with the analysis gridded fields, DIVA also provides error fields (Brankart and Brasseur, 1998;
Rixen et al., 2000) based on the data coverage and their noise. The method computes gridded fields in two dimensions. The 3D
(x,y,z) and 4D extensions (x,y,z,t) have been embedded into the interpolation scheme with emphasis in creating climatologies.
Detailed documentation of the method can be found in Troupin et al. (2010, 2012) and in the User Guide (Troupin et al., 2017).
The current Atlas has been produced with the use of diva-4.6.11 (Watelet et al., 2015a) with a Linux Ubuntu 16.04.3 operating
system.

### 3.2   Topography and Coastlines

The domain where the interpolation has to be performed is covered by a triangular finite-element mesh that follows the coast-
line. The bathymetry used for coastline definitions and mesh generation is based on the general Bathymetric Chart of the
Oceans (GEBCO) 1-min topography. Since the subsequent analysis focus on much larger scales than 1 min, the resolution of
the topography was downgraded to 5 minutes, still fine enough to resolve topological features such as islands and straits, yet
coarse enough to be coherent with the scales of interest. The step of the output grid was set to 1/8° for similar reasons. The
geographical boundaries of the region were set to 6.25°W-36.5°E, 30°-46°N. The depth contours that define the 2D horizontal
planes where the interpolation takes place are the 31 standard depth levels from 0 to 4000 m.





### 3.3 Finite Element Mesh

Once the depth contour files at the specified 31 standard depth levels were prepared, the mesh was generated using an initial correlation length $L_c$ equal to 0.5° which means the initial size $L_e$ of each finite triangular element is equal to 0.167° ($L_c/3$). The initial $L_c$ scale was taken as small as allowed by computing resources. This length scale is much smaller than the length

scales for the analysis (typically a few degrees) and it ensures that the finite element solution is solving the mathematical DIVA formulation with very high precision (it does not mean that the analysis is the truth, but that the numerical solution is actually close to the mathematical solution of the variational formulation; in other words, the discretization does not add further errors into the analysis than the analysis method and the data themselves). The mesh was generated once and used in each repetition of the analysis.

### 3.4 Interpolation Parameters

#### 3.4.1 Correlation length

The correlation length ($L_c$) is the radius of influence of data points. It can be determined objectively using a specific DIVA tool that takes into account the data distribution or provided a priori by the user according to its (subjective) experience on expected patterns occurring in the domain of interest. In the contrary to the SeaDataNet climatology where a constant value used in all

depths, in this Atlas, the correlation length was defined as follows. For the monthly climatology, the correlation length was calculated for every month (for all years from 1950 to 2015) and at each depth by fitting the empirical data covariance as a function of distance by its theoretical function. It was then smoothed by applying a vertical filtering, similarly to Troupin et al. (2010). Then, these monthly correlation length profiles were averaged together in order to have a smooth transition from one month to another. Indeed, the estimated correlation length at the surface vary between 1.8° (July) and 3.6° (December).

For the seasonal and the annual climatologies, the correlation length was calculated for every season and depth (from all years between 1950 and 2015), filtered vertically and then averaged into a single profile. This approach yields slightly smaller correlation lengths than the averaging per month and in combination with the rest analysis parameters, the fields are not so smoothed and are closer to the data. The mean profiles for temperature and salinity are shown in Fig. 3(a) and 3(c). There is a general increase from the surface to 2000 m except in a layer from 300 to 600 m where it decreases. Also there is a decrease

from around 2000 to 3000 m in both cases that can be attributed to the variability of the intermediate and deep waters in the Mediterranean.

#### 3.4.2 Signal-to-Noise Ratio

The signal-to-noise ratio (S/N) represents the ratio of the variance of the signal to the variance of observational errors. When measuring a variable, there is always an uncertainty on the value obtained. Noise does not only take into account instrumental errors (which are generally low), but it also includes:





(a) Monthly averaged correlation length

(b) Monthly averaged signal-to-noise ratio

(c) Seasonal averaged correlation length

(d) Seasonal averaged signal-to-noise ratio

**Figure 3.** Averaged profiles of correlation length (left) and signal-to noise ratio (right) for temperature and salinity.

1. the representativeness errors, meaning that what one measures is not always what one intends to analyse (e.g., skin temperature, inadequate scales, etc);



2. the synopticity errors, occurring when the measurements are assumed to be taken at the same time (e.g., data from a cruise) (Rixen et al., 2001).

Because of the multiple sources of error, a perfect fit to observations is thus not advised. In the case of climatology, the main cause of error is not the instrumental error but the representativeness errors, which cannot easily be quantified. In this Atlas, unlike the SeaDataNet and other regional climatologies (Troupin et al., 2010), a variable S/N was defined following the same approach as with the correlation length. The mean monthly and seasonal S/N profiles for temperature and salinity were vertically filtered. The remaining extremes were further filtered out manually by replacing them with the mean of the adjacent layers. The seasonal mean profile for temperature was further filtered out to a constant value from 1300 to 4000 m. In the first 500 m (see Figs 3(b) and 3(d)), both the monthly and seasonal profiles display similar behaviours with a mean value ranging from 3.5 to 4.5, with an exception of the monthly mean value for salinity at 400 m.

## 3.5  Background field

The background field is the first guess of the gridded field (analysis) to reconstruct. First of all, this background is subtracted from data. Then, the variational analysis is performed by DIVA using these data anomalies as an input. Lastly, the background is added to the solution so that original data and final analysis are both expressed as absolute values. In areas with very few or no data, the analysis tends to the background field since the data anomalies are close to zero (Brankart and Brasseur, 1998). In this Atlas, a semi-normed analysis field (roughly speaking an analysis only capturing very large scales) was chosen as the background in order to guarantee the solution is most realistic in areas void of data. Depending on the type of climatology and gridded fields two different background fields were used: (a) a climatic seasonal to account for seasonality from the original data, and (b) a climatic annual field to account for inter-annual variability. The same correlation lengths and signal-to-noise ratios were used as for the Atlas computations (see previous paragraph). Analytically the following background fields were produced:

1. Climatic seasonal semi-normed analysis as background for the decadal seasonal fields of temperature and salinity.

2. Climatic annual semi-normed analysis as background for the decadal annual fields of temperature and salinity.

3. Climatic seasonal semi-normed analysis as background for the periodical seasonal fields of temperature and salinity.

4. Climatic annual semi-normed analysis as background for the periodical annual fields of temperature and salinity.

5. Climatic seasonal semi-normed analysis as background for the monthly, seasonal climatology.

6. Data mean value as background for the annual climatologies.

Additional monthly and seasonal climatologies were computed using the data mean value as background (the data mean value is subtracted from the data values).

The background climatic seasonal and annual fields were detrended in order to remove any seasonal and inter-annual trends
5   that are due to non-uniform spatial data distribution (Capet et al., 2014). A characteristic example is the non-uniform distribution of thermosalinographs. Also, the monthly, seasonal and annual climatologies were detrended. A future target is to apply the detrending for the estimation of the biases induced to the temperature data due to instrumental errors such as the bathythermographs.

## 4   Climatological gridded fields

The content of the Atlas is described in Section 1. For each gridded error field, maps are provided that allow one to assess the reliability of the gridded fields and to objectively identify areas with poor coverage. DIVA provides two different errors, the relative (from 0 to 1) and the absolute errors (expressed in physical parameter units), which depend on the accuracy of the
observations and their distribution. The relative error has been chosen in this atlas and two threshold values are used (equal to 0.3 or 0.5) for the quality assessment of the results (Fig. 4).

### 4.1   Description of the gridded fields

While the physical interpretation of the maps is not in the scope of this paper, it is worth noting that the main physical processes are well resolved both in space and time, for instance:

1. The clear signature of the temperature and salinity gradient from west to east, shown in all monthly, seasonal and annual plots at all depths.

2. The temperature and its seasonal cycle.

3. The Rhodes gyre with the low temperatures is evident in all plots of temperature at surface layers regardless month or season.

4. The Black Sea outflow with the low temperature and salinity values, which is evident in the surface plots, regardless month or season.

Figure 5 illustrates the surface temperature at 5 m in January, April, July and October and represents the seasonal cycle of the thermal field with a mean shift of 9°C between January and July. The Western Mediterranean exhibits smaller temperatures than the Eastern basin. The differences between the northern and southern regions (meridional gradient) are also very well
represented. The Rhodes Gyre, identified by minimum temperature values is distinguished southeast of Crete island, mainly at spring and autumn periods.

Some representative seasonal distributions of salinity at 5 m are shown at Fig. 6. The signal of the fresh Atlantic water (AW) (S ≈ 36.5) and its flow along the Algerian coast towards the eastern basin through the Sicily Strait is very evident. We can also notice the areas of low salinity due to river runoffs such as the North Adriatic and the North Aegean Sea that is influenced by
the Dardanelles outflow.

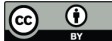



**Figure 4.** Example of observations distribution (a) and corresponding fields distributions depending on relative error threshold values, (b) unmasked field, (c) masked field with relative error threshold=0.3, (d) masked field with relative error threshold=0.5.

Winter salinity fields at 100 m and 500 m respectively are shown at Figs. 7 and 8 for the last four periods (1970–1979, 1980–1989, 1990–1999, and 2000–2015). There is a homogeneous increasing trend from west to east compared to the surface. The salinity range is between about 37 and 38.6 at 100 m at the western Mediterranean while the periods 1980–1989 and 2000–2015 are characterized by higher salinities compared to the other periods. Maximum values are found at the south Adriatic at 38.9 the period 1980–1989 and 38.8 the period 2000–2015. At the Eastern Mediterranean, the salinity is between about 38.4 and 39.4 at 100 m with the maxima found in the Levantine basin and Aegean Sea. The last period 2000–2015 is at an average more saline than the previous ones and the thermohaline circulation patterns (such as the Shikmona Gyre) are more intense.

In the western basin it is characteristic the increase of salinity at 500m compared to the 100m, where it is between about 38.4 and 38.7. This increase is due to the mixing with LIW that flows towards the Strait of Gibraltar. The Tyrrhenian Sea has higher salinities in the periods 1970–1979 and 2000–2015. In the Eastern Mediterranean the salinity is between about 38.4 and



**Figure 5.** Surface temperature climatology at 5 m in (a) January, (b) April, (c) July, and (d) October.

39.2 with the highest values at the North and South Aegean Sea and the period 2000–2015 be more saline than the previous years.

## 4.2 Comparison with SeaDataNet climatology

For the current Atlas, similar methodology as SeaDataNet was used, allowing a direct comparison between both products. The SeaDataNet climatology is defined between 9.25°W–36.5°E of longitude and 30–46°N of latitude with an horizontal resolution of 1/8° × 1/8° on 30 IODE vertical standard levels from 0 to 4000m. DIVA software version 4.6.9 (Watelet et al., 2015b) has been used for both the analysis and background field computation. The salinity background field has been computed through annual semi-normed analysis considering all available observations, while for temperature, 3–months semi–normed background fields centered on the analysis month have been considered due to the large temperature seasonal variability.





**Figure 6.** Surface salinity climatology at 5 m for (a) Winter, (b) Spring, (c) Summer and (d) Autumn.

Figure 9 shows temperature distributions for Atlas and SeaDataNet climatology, for January at 10 m and November at 1000 m. There is very good consistency between both products. The same good agreement exists among other months and depths (not shown here). In surface distributions (at 10 m), the data weighting applied in the Atlas (Fig. 9(a)) has reduced the extension of the influence of Po river.

Statistical indexes like bias (differences between the two climatologies) and RMSE (Root-Mean-Square Error) show very good agreement too. Figure 10 shows the basin vertical averages of bias and RMSE between Atlas and SeaDataNet monthly temperature (a) and monthly salinity (b) climatologies. Higher differences occur in spring and autumn, Atlas is 0.03°C more warm than SeaDataNet in June and less warm at 0.02°C in November. Temperature RMSE values are ranging between 0.09 in February and 0.19 in July. Maximum differences between the two climatologies are located in the first layers (not shown here) for all months. Atlas has slightly higher than SeaDataNet salinity values at August and December and the mean bias





**Figure 7.** Winter salinity at 100 m for four periods: (a) 1970–1979, (b) 1980–1989, (c) 1990–1999, (d) 2000–2015.

value throughout the year is 0.003. Salinity RMSE values are ranging between 0.048 and 0.076. As for temperature, maximum differences are located in the first layers for al months.

5    ## 4.3    Comparison with World Ocean Atlas (WOA13)

WOA13 provides objectively analyzed annual, seasonal and monthly fields covering the time period 1955–2012 (average of six decadal means), and six decades: 1955–1964, 1965–1974, 1975–1984, 1985–1994, 1995–2004, 2005–2012. Each decadal climatology consists of annual (computed as 12-months averages); seasonal: Winter (Jan–Mar), Spring (Apr–Jun), Summer (Jul–Sep), Fall (Oct–Dec) computed as 3-month averages; and monthly fields (above 1500 m). Annual, seasonal and monthly

10    temperature and salinity fields are available on $1° × 1°$ and $1/4° × 1/4°$ latitude/longitude grids. All annual and seasonal fields were calculated from 0 to 5500 m depth on 102 standard levels. Monthly fields are available only above 1500 m on all grids





**Figure 8.** Winter salinity at 500 m for four periods: (a) 1970–1979, (b) 1980–1989, (c) 1990–1999, (d) 2000–2015.

on 57 standard levels. Figure 11 shows temperature fields for Atlas and WOA13 for January at 0 m and September 1000 m. At the surface, the Atlas is able to represent the January mean state with more details.

Statistical indexes to quantify the differences between several vertical averages have been calculated also which show a
5  very good agreement between both monthly T/S climatologies. Figure 12 shows the basin vertical averages of bias and RMSE between Atlas and WOA13 monthly temperature (a) and monthly salinity (b) climatologies. Temperature bias is always positive indicating that the Atlas is slightly warmer than the WOA13 climatology. Atlas salinity is less than the WOA13 at an average of 0.007 throughout the whole year.



**Figure 9.** Comparison of Atlas temperature (left) and SeaDataNet climatology (right) between January at 10 m (top) and November at 1000 m (bottom).

## 4.4 Comparison with Medar/Medatlas Climatology

The Medar/Medatlas climatology was computed using the Variational Inverse Method (VIM) as the current Atlas. Correlation length was fixed a priori to a constant value according to the a priori knowledge of typical scales of parameters in the domain of interest. The signal-to-noise ratio was calibrated by a Generalized Cross Validation technique. Data closer than 15 km from the coast or in areas shallower than 50 m were rejected in order to avoid the influence of coastal features on open sea (Rixen et al., 2005).

The impact of such as rejection can be seen in the Fig. 13: in the North Adriatic Sea and in the North Aegean Sea, the influence of Po river and Black Sea outflows during winter are not captured.





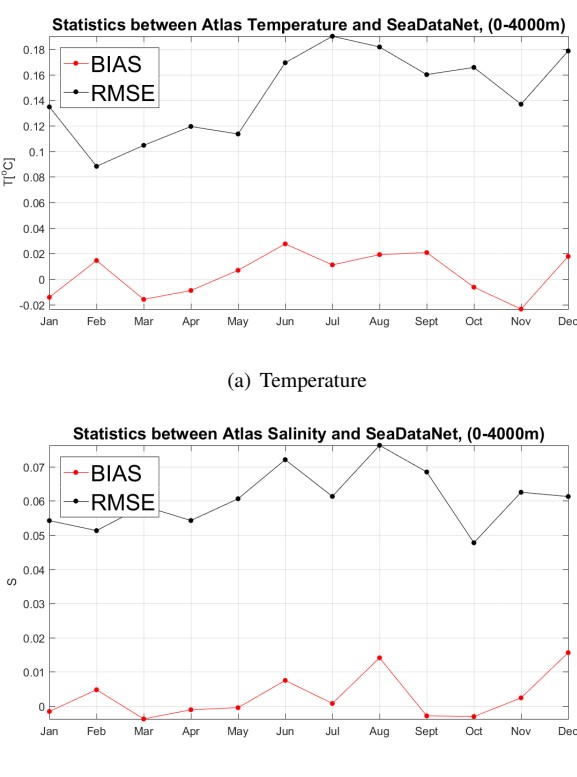

(a) Temperature

(b) Salinity

**Figure 10.** Basin vertical averages of BIAS and RMSE between Atlas and SeaDataNet monthly temperature (a), and monthly salinity (b) climatologies.

The temperature at 10 m for the decade 1991–2000 is shown below for the Atlas (Fig. 14(a)) and the Medar/Medatlas climatology (Fig. 14(b)). The Medar/Medatlas profiles extend until 2000, so this decade was chosen as the most complete one since there is no error field available to mask regions empty of data.

As it can be seen in Fig. 14, the current Atlas describes with more detail the several features of the general circulation pattern.

5  # 5  Conclusions

A new, high-resolution Atlas of temperature and salinity for the Mediterranean Sea for the period 1950–2015 is presented. The analysis is based on the latest SeaDataNet dataset, providing the most complete, extended and improved collection of in-situ observations.

The results describe well the expected distributions of the hydrological characteristics and reveal their long term changes. 10  Techniques for overcoming the inhomogeneous vertical and spatial distribution of oceanographic measurements were presented. The analysis focused on data of 1950 and onwards where more reliable instrumental observations exist. Comparisons with SeaDataNet reveal a very good agreement. It was expected since both two climatologies use similar interpolation method-





**Figure 11.** Comparison of Atlas temperature (left) and WOA13 climatology (right) between January at 0 m (top) and September at 1000 m (bottom).

ology. The Atlas offers additional value-added decadal temperature and salinity distributions which already exist in the region from previous versions of Medar/Medatlas climatology though with no error fields available. This is of particular importance.

There are still improvements to be implemented in future versions of the atlas, such as a three-dimensional analysis instead of stacking the two-dimensional horizontal levels together in order to take into account the vertical correlation of the parameters. This would reduce vertical inconsistencies that may remain in the results. In addition, the correction of the warm bias in the bathythermographs data, caused by instrumental errors, should also be addressed. However it is anticipated that the approach followed here concerning the calibration of the analysis parameters will be followed by other groups in the future for the Mediterranean climate studies and other applications related with the long term variability of the hydrological characteristics of the region and its climate change.



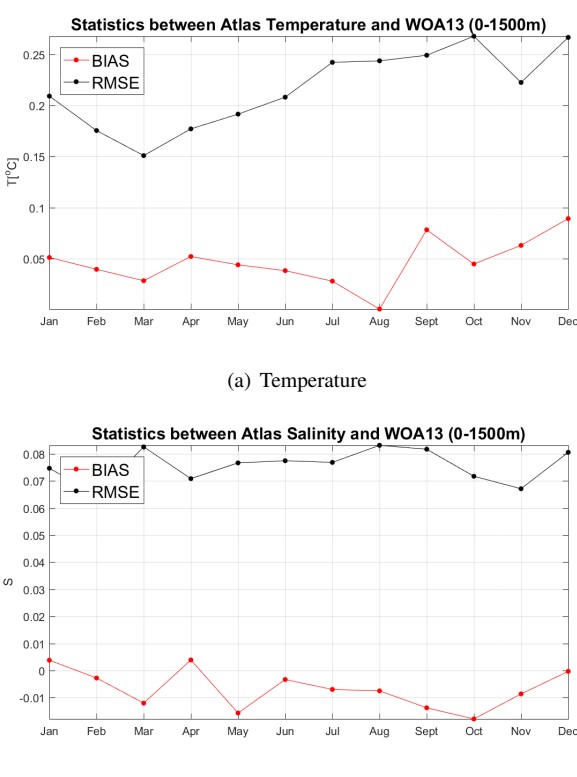

(a) Temperature

(b) Salinity

**Figure 12.** Basin vertical averages of BIAS and RMSE between Atlas and WOA13 monthly temperature (a), and monthly salinity (b) climatologies.

Another future improvement is the use of a dynamical constraint such as a real velocity field or an advection constraint based on topography that would allow anisotropies in the correlations to be included into the analysis. This Atlas aims to contribute

5    to existing available knowledge in the region and fill existing data gaps in space and time.

## 6   Code and data availability

The dataset used in this work is available from https://doi.org/10.12770/8c3bd19b-9687-429c-a232-48b10478581c. Updated versions will be released periodically.

The DIVA source code is distributed via GitHub at https://github.com/gher-ulg/DIVA. The different versions of the software

10   tool are archived and referenced in Zenodo platform under the DOI: https://doi.org/10.5281/zenodo.592476.

The Atlas itself is distributed through Zenodo according to the following sub-products:

**Annual Climatology:**  https://doi.org/10.5281/zenodo.1146976.



**Figure 13.** January temperature at 10 m: (a) Medar/Medatlas, coastal data removed from the analysis; (b) data used; (c) current climatology, coastal data included; (d) data used.

**Seasonal Climatology for 57 running decades** from 1950-1959 to 2006-2015:
https://doi.org/10.5281/zenodo.1146938.

5   **Seasonal Climatology:** https://doi.org/10.5281/zenodo.1146953.

**Annual Climatology for 57 running decades** from 1950-1959 to 2006-2015:
https://doi.org/10.5281/zenodo.1146957.

**Seasonal Climatology for six periods:** 1950-1959, 1960-1969, 1970-1979, 1980-1989, 1990-1999, 2000-2015:
https://doi.org/10.5281/zenodo.1146966.





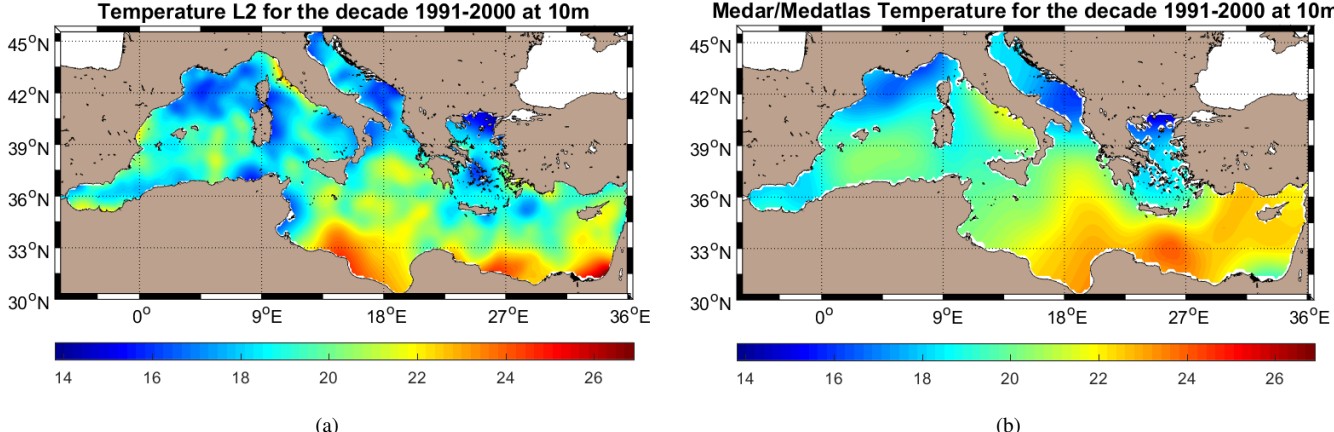

**Figure 14.** (a) Atlas masked temperature field for the decade 1991–2000 at 10 m, (b) Medar/Medatlas temperature field for the decade 1991–2000, at 10 m (no error field is available).

**Annual Climatology for six periods:** 1950-1959, 1960-1969, 1970-1979, 1980-1989, 1990-1999, 2000-2015:

https://doi.org/10.5281/zenodo.1146970.

**Monthly Climatology:** https://doi.org/10.5281/zenodo.1146974.

*Author contributions.* A.I. created the Atlas, wrote the first version of the manuscript and prepared the figures. JMB, S.W., C.T. and A.T.
5  reviewed the manuscript. C.T. and S.W. formatted the document in LaTeX.

*Competing interests.* The authors declare no competing interests.

*Disclaimer.* It cannot be warranted that the Atlas is free from errors or omissions. Correct and appropriate Atlas interpretation and usage is solely the responsibility of data users.

*Acknowledgements.* The DIVA development has received funding from the European Union Sixth Framework Programme (FP6/2002–2006)
10  under grant agreement no. 026212, SeaDataNet, Seventh Framework Programme (FP7/2007–2013) under grant agreement no. 283607,
SeaDataNet II, SeaDataCloud and EMODnet (MARE/2008/03 - Lot 3 Chemistry - SI2.531432) from the Directorate-General for Maritime
Affairs and Fisheries.



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
