# Peer review of "Mediterranean Sea Hydrographic Atlas: towards optimal data analysis by including time-dependent statistical parameters"

_Earth System Science Data, 2018_

## Referee Comment (RC1) · Anonymous Referee #1 · 6 Feb 2018

General Comments. The paper is presenting a new climatological atlas of the Mediterranean Sea and discussing its quality. The authors are well presenting the problems related to different data collected with different technologies and to the uneven distribution of data. The methodology applied to calculate the climatologies is particularly useful in these cases. The Mediterranean is a semi-enclosed concentration basin with dense and deep water formation. The reduced Rossby radius and the high mesoscale variability should be discussed: how do they influence the data variability. The selection of the grid size is some sense is also filtering out some phenomena. A short discussion on a such effect would be beneficial for the future use of the climatologies.

[Figure]

Specific comments. In the abstract it is cited the original source of data (https://doi.org/10.12770/8c3bd19b-9687-429c-a232-48b10478581c) and not the climatologies produced by the authors: this should be changed. Furthermore, data source is accessible only through a password. Indications on how to access them must be provided.

In the abstract a sentence should have an additional information (lines 17-18): especially in critical areas of interest such as the Marine Strategy Framework Directive (MSFD) regions AND SUBREGIONS.

In the Introduction the physical characteristics of the Mediterranean water masses are presented: the authors should specify the unit of the salinity and, if it is in absolute salinity, how the old values have been transformed.

line 20, instead of Temperature and Salinity values, for the Cretean Sea waters the density has been provided: to be coherent, also in this case T and S should be provided.

line 33 it is mentioned a V1.1 version of the Simoncelli climatology: it is V2. Correct the value also in other places.

---

## Author Comment (AC1) · 10 Feb 2018

The authors would like to thank the reviewer for the comments and suggestions for the improvement of the manuscript.

Please see below is *our response (in italics)* and the *changes (in red)* following the **reviewer's comments (in bold).**

Athanasia Iona, on behalf of the authors' team.

**Comments by Referee #1**

**General Comments. The paper is presenting a new climatological atlas of the Mediterranean Sea and discussing its quality. The authors are well presenting the problems related to different data collected with different technologies and to the uneven distribution of data. The methodology applied to calculate the climatologies is particularly useful in these cases. The Mediterranean is a semi-enclosed concentration basin with dense and deep water formation. The reduced Rossby radius and the high mesoscale variability should be discussed: how do they influence the data variability. The selection of the grid size is some sense is also filtering out some phenomena. A short discussion on a such effect would be beneficial for the future use of the climatologies.**

> *Reply to the reviewer:*

> *The climatological analysis of large historical data collections using the variational methodology is not very sensitive to the correlation length scales (Brankart J.M. and Brasseur P, 1997) since the spatial data coverage is such that the corresponding information is already contained in the data and the data variability is empirically taken into account when optimizing the correlation lengths.*

> *The reviewer is of course right that mesoscale features are filtered out. However it is not primarily the spatial grid but the time averaging used to define the analysis periods which filter out those processes.*

> *To clarify we will add the following text:*

> *The climatologies provided cannot rely on sufficient high frequency and high resolution data to allow resolving mesoscale features which play an important role and modify the large scale flow fields (Robinson et al, 2001). We focus thus on the seasonal and decadal variations. This time filtering also results in a spatial filter as later shown by the spatial correlations found in the data. The spatial scales the data can capture are of the order of 300-350 km at the surface, much larger than the Rossby radius of deformation scale (10-15 km) associated with mesoscale motions (30-80 km, Robinson et al, 2001). These mesoscale features are thus filtered out from the analysis and hence the numerical grids we will use only needs to resolve the large scales. The same holds for the output files, where there is no reason to save at very high resolution (much smaller than the deformation radius) as in any case the analysis provides large scale fields.*

**Specific comments. In the abstract it is cited the original source of data (https://doi.org/10.12770/8c3bd19b-9687-429c-a232-48b10478581c) and not the climatologies produced by the authors: this should be changed. Furthermore, data source is accessible only through a password. Indications on how to access them must be provided**.

*Reply to the reviewer:*

*In the revised manuscript, at the abstract, next to the original data source citation the links for the access of the climatologies will be added, namely: "The climatologies in netCDF are available at:*
*Annual Climatology: https://doi.org/10.5281/zenodo.1146976;*
*Seasonal Climatology for 57 running decades: https://doi.org/10.5281/zenodo.1146938;*
*Seasonal Climatology: https://doi.org/10.5281/zenodo.1146953;*
*Annual Climatology for 57 running decades: https://doi.org/10.5281/zenodo.1146957;*
*Seasonal Climatology for six periods: https://doi.org/10.5281/zenodo.1146966;*
*Annual Climatology for six periods: https://doi.org/10.5281/zenodo.1146970;*
*Monthly Climatology: https://doi.org/10.5281/zenodo.1146974."*

*Concerning the indications on how to access the data source, in section 2.1, at the end of the first paragraph it will be added: "Users have to register in the Marine-ID (https://users.marine-id.org) to get an account for downloading the SeaDataNet V2 data collection. Registration is done only once and thereafter users can have access not only to SeaDataNet but all EMODnet and Copernicus marine data services."*

**In the abstract a sentence should have an additional information (lines 17-18): especially in critical areas of interest such as the Marine Strategy Framework Directive (MSFD) regions AND SUBREGIONS.**

*Reply to the reviewer:*

*In the revised manuscript it will be added the additional information "and subregions." as requested.*

**In the Introduction the physical characteristics of the Mediterranean water masses are presented: the authors should specify the unit of the salinity and, if it is in absolute salinity, how the old values have been transformed.**

*Reply to the reviewer:*

*The unit of the salinity is in ppt. In the revised version of the manuscript, the unit of the salinity (ppt) will be added where it is appropriate.*

**line 20, instead of Temperature and Salinity values, for the Cretean Sea waters the density has been provided: to be coherent, also in this case T and S should be provided.**

*Reply to the reviewer:*

*In the revised manuscript the mean density value there will be replaced by the mean temperature and salinity values namely "S~39 psu, T~14.8 $^{o}$C"*

**line 33 it is mentioned a V1.1 version of the Simoncelli climatology: it is V2. Correct the value also in other places.**

*Reply to the reviewer:*

*We are sorry, there is an error at the link provided at line 33 of the manuscript while the mentioned version V1.1 is correct. The SeaDataNet climatology is based on V1.1 collection while this Atlas is using the newer version V2 which includes more data. The correct link to be included at line 33 of the revised manuscript is "http://doi.org/10.12770/cd552057-b604-4004-b838-a4f73cc98fcf"*

---

## Referee Comment (RC2) · Anonymous Referee #1 · 19 Feb 2018

the authors provided all the requested integrations and updated the paper accordingly. No more integrations are required and the paper is ready for publication.

---

## Referee Comment (RC3) · A. Mishonov (Referee) · 1 Mar 2018

This ms is describing a very important step in developing of the comprehensive decadal regional climatology for the Mediterranean Sea - a very important region of the world ocean. This climatology provide an improved oceanographic foundation and reference for multi-disciplinary studies with its high-resolution quality-controlled temperature and salinity fields on standard depth levels from the sea surface to 5,500 m depth. The individual decadal fields are a significant upgrade from the previous version of the Mediterranean Sea climatology. This could be quite useful for assessing regional climate change over the long time period and can be utilized in other climate-related

applications. Using DIVA tools for data processing is interesting approach and it is appears to be quite successful. I will be very much interested to see how this Atlas will be used in climate change research in that region of the world ocean.

I believe this ms can be published after applying several minor technical corrections listed below.

——My answers to the standard questions: 1. Are the data and methods presented new? Historical data processed using renewed method

2. Is there any potential of the data being useful in the future? Yes, absolutely

3. Are methods and materials described in sufficient detail? Yes

4. Are any references/citations to other data sets or articles missing or inappropriate? No

5. Is the article itself appropriate to support the publication of a data set? Yes

6. Is the data set accessible via the given identifier? Yes

7. Is the data set complete? Yes

8. Are error estimates and sources of errors given (and discussed in the article)? Yes

9. Are the accuracy, calibration, processing, etc. state of the art? Yes

10. Are common standards used for comparison? Yes

11. Is the data set significant – unique, useful, and complete? Yes

12. Are there any inconsistencies within these, implausible assertions or data, or noticeable problems which would suggest the data are erroneous (or worse). If possible, apply tests (e.g. statistics). Unusual formats or other circumstances which impede such tests in your discipline may raise suspicion. No

13. Is the data set itself of high quality? Yes
14. Is the data set usable in its current format and size? Yes

15. Are the formal metadata appropriate? Yes

16. Is the length of the article appropriate? Yes, the ms is long, but it is appropriate.

17. Is the overall structure of the article well structured and clear? Yes

18. Is the language consistent and precise? Yes

19. Are mathematical formulae, symbols, abbreviations, and units correctly defined and used? Yes

20. Are figures and tables correct and of high quality? Yes (see remarks about Figs and Table 1 below)

21. Is the data set publication, as submitted, of high quality? Yes

22. By reading the article and downloading the data set, would you be able to understand and (re-)use the data set in the future Yes, I've downloaded all data using links provided and was able to work with the Atlas.

——— Remarks:

1. Page 4, Line 25 in 1.1 Objectives chapter: word 'fields' is missing in ' Seasonal climatological gridded obtained by analyzing. . .' sentence.

2. Page 4, Line 32 in 1.1 Objectives chapter: word 'obtained' is missing in ' Annual gridded fields by analyzing all. . .' sentence.

3. Page 5. Part 2 Data, Line 21: correct INVG to INGV.

4. Page 6: Table 1 - remark 1: WOA13 listed as having 57 levels for all temporal resolutions. In WOA13 57 levels are only for monthly fields. Annual and seasonal fields made on 102 levels. - remark 2: WOA13 based on WOD13, which consist of data collected from following platforms: Ocean Station Data – OSD; High-resolution Conductivity-Temperature-Depth – CTD; Mechanical/Digital/Micro Bathythermograph

– MBT; Expendable Bathythermograph – XBT; Surface – SUR; Autonomous Pinniped Bathythermograph – APB; Moored Buoy – MRB; Profiling Float – PFL; Drifting Buoy – DRB; Undulating Oceanographic Recorder – UOR; and Glider – GLD. - remark 3: WOA13 consist of several parameters, not only T & S: Temperature (°C) Salinity (unitless) Density (kg/m3) beta version Conductivity (S/m) Dissolved Oxygen (ml/l) Percent Oxygen Saturation (%) Apparent Oxygen Utilization (ml/l) Silicate ($\mu$mol/l) Phosphate ($\mu$mol/l) Nitrate ($\mu$mol/l) - remark 4: In addition to the listed Temporal resolution, WOA13 consist of several decadal climatologies (that is correctly stated in 1.1 Objectives chapter, lines 5-10).

Please correct info in Table (all correct information is presented on page 19, part 4.3, lines 6-11).

5. Page 7: Figure 1 - could be made bigger.

6. Page 11, Part 3.1 The Diva..., Line 11: replace 'filed' with 'field' in '..a continuous filed approximating...'

7. Page 13, Fig. 3: would be good to have a titles/units for X&Y-axis.

8. Page 16, Fig 4: I would suggest to keep the T range for (b) plot similar to (c), & (d) plots.

9. Page 17, Fig.5: I would suggest to keep the same T range for a-d plots (let say 6-28C), so they will be visually comparable (as it is done for S on Figs 6, 7, & 8).

10. Page 25, Fig 13: top-aligning the a-b & c-d plots will improve the appearance of this figure. ——

---

## Author Comment (AC2) · 4 Apr 2018

The authors would like to thank the referee A. Mishonov  for his constructive and detailed review, his remarks on technical corrections for the improvement of the manuscript and the positive comments about the importance of the Atlas for accessing regional climate change over the long time period.

Please see below is *our response (in italics)* and the changes (in red) following the **referee review and remarks for minor technical corrections (in bold).**

**Review by Referee #2**

**This ms is describing a very important step in developing of the comprehensive regional climatology for the Mediterranean Sea - a very important region of the world ocean. This climatology provide an improved oceanographic foundation and reference for multi-disciplinary studies with its high-resolution quality-controlled temperature and salinity fields on standard depth levels from the sea surface to 5,500 m depth. The individual decadal fields are a significant upgrade from the previous version of the Mediterranean Sea climatology.  This could be quite useful for assessing regional climate change over the long time period and can be utilized in other climate-related applications. Using DIVA tools for data processing is interesting approach and it is appears to be quite successful. I will be very much interested to see how this Atlas will be used in climate change research in that region of the world ocean.**

*General comment to the referee review:*

> *We would like to inform the referee that a direct output of the Atlas (and the subject of a second paper) is a set of climatic indices such as long time series of temperature, salinity, ocean heat/salt content anomalies, differences of spatial patterns of anomalies between decades etc for accessing the climate change regime of the Mediterranean basin.*

**Reviewer's remarkrs for minor technical corrections.**

1.  **Page 4, Line 25 in 1.1 Objectives chapter:  word 'fields' is missing in ' Seasonal climatological gridded obtained by analyzing ... ' sentence.**

    *Reply to the reviewer:*
    *In the revised manuscript, the word 'fields' will be added and the sentence will become:*
    *"Seasonal climatological gridded fields obtained by analyzing all data of the whole period from 1950 to 2015 falling within each season."*

2.  **Page 4, Line 32 in 1.1 Objectives chapter:  word 'obtained' is missing in ' Annual gridded fields by analyzing all ...' sentence.**

    *Reply to the reviewer:*

    *In the revised manuscript, the word 'obtained' will be added and the sentence will become:*
    *"Annual gridded fields obtained by analyzing all data regardless month or season for each of the 57 running decade from 1950–1959 to 2006–2015."*

3. **Page 5, Part 2 Data, Line 21: correct INVG to INGV.**

   *Reply to the reviewer:*
   *In the revised manuscript INVG will be corrected to "INGV".*

4. **Page 6, Table 1 - remark 1: WOA13 listed as having 57 levels for all temporal resolutions. In WOA13 57 levels are only for monthly fields. Annual and seasonal fields made on 102 levels. - remark 2: WOA13 based on WOD13, which consist of data collected from following platforms: Ocean Station Data – OSD; High-resolution Conductivity-Temperature-Depth – CTD; Mechanical/Digital/Micro Bathythermograph – MBT; Expendable Bathythermograph – XBT; Surface – SUR; Autonomous Pinniped Bathythermograph – APB; Moored Buoy – MRB; Profiling Float – PFL; Drifting Buoy – DRB; Undulating Oceanographic Recorder – UOR; and Glider – GLD. - remark 3: WOA13 consist of several parameters, not only T & S: Temperature (?C) Salinity (unitless) Density (kg/m3) beta version Conductivity (S/m) Dissolved Oxygen (ml/l) Percent Oxygen Saturation (%) Apparent Oxygen Utilization (ml/l) Silicate (μmol/l) Phosphate (μmol/l) Nitrate (μmol/l) - remark 4: In addition to the listed Temporal resolution, WOA13 consist of several decadal climatologies (that is correctly stated in 1.1 Objectives chapter, lines 5-10).**

   **Please correct info in Table (all correct information is presented on page 19, part 4.3, lines 6-11).**

   *Reply to the reviewer:*
   *The reviewer is right that there is missing information about WOA13 in Table1. The content of Table 1 was intended to describe the available climatologies in Mediterranean that would be compared with the new Atlas and in case of WOA13, the T,S climatology on $1/4^o \times 1/4^o$ horizontal resolution, at 57 leveles, from 0-1500m, but this is not clearly stated neither in the Table 1 caption neither in the main text. In addition, in case of Medar/Medatlas there are mentioned additional than the T,S parameters.*
   *Therefore, all the remarks will be addressed. Analytically:*
   *Remark 1: in Table1, at the corresponding cell it will be added "57 levels for monthly fields, 102 levels for annual, seasonal fields."*

   *Remark 2: in Table1, at the corresponding cell it will be added analytically all instruments/platforms names, e.g. "Ocean Station Data – OSD; High-resolution Conductivity-Temperature-Depth – CTD; Mechanical/Digital/Micro Bathythermograph – MBT; Expendable Bathythermograph – XBT; Surface – SUR; Autonomous Pinniped Bathythermograph – APB; Moored Buoy – MRB; Profiling Float – PFL; Drifting Buoy – DRB; Undulating Oceanographic Recorder – UOR; and Glider – GLD."*

   *Remark 3: In addition to Temperature ($^o$C) and Salinity (unitless) all the rest climatologies will be added in Table 1 e.g "Density (kg/m3) beta version Conductivity (S/m) Dissolved Oxygen (ml/l) Percent Oxygen Saturation (%) Apparent Oxygen Utilization (ml/l) Silicate (μmol/l) Phosphate (μmol/l) Nitrate (μmol/l)."*

   *Remark 4: in the temporal resolution of WOA13 the decadal climatologies it will be added as: "Climatic, monthly, seasonal, averaged periods, decadal (1955-64, 1965-74, 1975-84, 1985-94, 1995-2004, 2005-2012 years)."*

5. **Page 7, Figure 1 - could be made bigger.**

*Reply to the reviewer:*
*Figure 1 is of high resolution and allows the modification of size. In the revised manuscript Figure 1 will be enlarged to the page width.*

6. **Page 11, Part 3.1 The Diva ..., Line 11: replace 'filed' with 'field' in '..a continuous filed approximating...'**

*Reply to the reviewer:*
*In the revised manuscript the "filed" will be replace with "field".*

7. **Page 13, Fig. 3: would be good to have a titles/units for X&Y-axis.**

*Reply to the reviewer:*
*In the revised manuscript the titles/units for X&Y will be added as in the figure below ("Depth (m) for Y axis", "Monthly-Seasonal Correlation Length (degrees)", and "Monthly-Seasonal Signal-to-Noise Ratio").*

[Figure]

[Figure]

8. **Page 16, Fig. 4: I would suggest to keep the T range for (b) plot similar to (c), & (d) plots.**

*Reply to the reviewer:*
*In the revised manuscript the following image with the corrected T range will replace Figure 4b:*

[Figure]

9. **Page 17, Fig. 5: I would suggest to keep the same T range for a-d plots (let say 6-28C), so they will be visually comparable (as it is done for S on Figs 6, 7, & 8).**

   *Reply to the reviewer:*
   *The temperature seasonal cycle is more marked than the salinity's with high differences between winter and summer (mean shift about $9^oC$). The T range was intentionally changed from one month to another because the focus was not to show the seasonal cycle but the T gradients from north to south and from east to west. However, as this is not explicitly explained in the manuscript the reviewer's suggestion is correct and for consistency reasons with the other figures,* the T range will change and be kept constant as below to allow the visual comparison:

[Figure]

   *(a)*                                                    *(b)*

   *(c)*                                                    *(d)*
   *Figure 5: Surface temperature climatology at 5 m in (a) January, (b) April, (c) July, and (d) October.*

10. **Page 25, Fig. 13: top-aligning the a-b & c-d plots will improve the appearance of this figure.**

    *Reply to the reviewer:*
    In the revised manuscript the vertical alignment of  a-b-c-d plots will be set to top.